# Patient-Centered Therapy for Obstructive Sleep Apnea: A Review

**DOI:** 10.3390/medicina58101338

**Published:** 2022-09-23

**Authors:** Pahnwat Taweesedt, Hala Najeeb, Salim Surani

**Affiliations:** 1Corpus Christi Medical Center, Corpus Christi, TX 78411, USA; 2Dow Medical College, Dow University of Health Science, Karachi 74200, Pakistan; 3Health Science Center, Texas A&M University, College Station, TX 79016, USA

**Keywords:** patient-centered, obstructive sleep apnea, patient preference, phenotypes, CPAP, OSA, personalized care, personalized medicine

## Abstract

Obstructive sleep apnea (OSA) is one of the most common sleep problems defined by cessation or decreased airflow despite breathing efforts. It is known to be related to multiple adverse health consequences. Positive airway pressure (PAP) is considered an effective treatment that is widely used. Various modes of PAP and other emerging treatment options are now available. A multidisciplinary approach, understanding diverse phenotypes of OSA, and shared decision-making are necessary for successful OSA treatment. Patient-centered care is an essential modality to support patient care that can be utilized in patients with OSA to help improve outcomes, treatment adherence, and patient satisfaction.

## 1. Introduction

Obstructive sleep apnea (OSA) is one of the most common sleep problems. OSA prevalence in the general population ranges from 9–38% [1], with a significant economic burden with undiagnosed sleep apnea costing USD 149.6 billion in the United States. This includes USD 89.6 in lost productivity and USD 26.2 billion in vehicular motor accidents (MVA). Patients with untreated OSA have a 243% higher risk of MVA than individuals without OSA [2].

The prevalence of OSA continues to increase substantially, likely related to the rising obesity prevalence. OSA can occur in both genders and increases in advanced age [3]. OSA is characterized by a decreased or complete pause in airflow despite the effort to breathe during sleep, resulting in intermittent hypoxemia. The diagnosis of OSA can be made either at home with home-sleep apnea testing or in-laboratory overnight sleep testing with in-laboratory polysomnography (PSG). The latter test provides more details, but is more expensive, time-consuming, and requires a sleep technician to monitor. Common complaints among patients with OSA include excessive daytime sleepiness, nocturia, fatigue, witness apnea, and morning headaches [4]. The cause of OSA has been reported to be muscle relaxation during sleep, leading to repetitive upper airway collapse and decreased blood oxygenation.

Several risk factors have been reported to be associated with OSA, for example, older age, male gender, obesity, large neck size, and craniofacial morphology. Various adverse outcomes are associated with OSA, including motor vehicle accidents, metabolic syndrome, neuropsychiatric dysfunction, cancer, cardiovascular diseases, cerebrovascular morbidity, and mortality [5]. Without treatment, OSA can cause a substantial economic burden on the healthcare system [6].

Understanding pathophysiology and different OSA endophenotypes will facilitate selecting the proper treatment. A structural or physiological pathways approach can be taken in managing patients with OSA. Structural or anatomical impairment in OSA includes the main risk factors, obesity and craniofacial anomalies, such as bilateral mandibular hypoplasia and craniofacial microsomia [7]. Obesity and a large neck circumference are well-known OSA risks because of an increase in neck adipose tissue leading to a high propensity of pharyngeal collapse [8]. Muscles that can lead to airway collapse include (1) tongue muscles (genioglossus), (2) muscles affecting the position of the hyoid bone (sternohyoid and geniohyoid), (3) muscles of the soft palates (levator veli palatini and tensor veli palatini) [9]. Patient evaluation is, therefore, essential to identify these abnormalities. In OSA populations with anatomical problems, intensive lifestyle modifications, hypoglossal nerve electrical stimulation, myofunctional therapy, oral appliance therapy, pharmacological treatment, and surgical correction such as a tonsillectomy or bariatric surgery should be considered [10]. On many occasions, OSA is classified based on its physiological phenotypes, including collapsibility, over loop gain (instability of ventilatory control system), low and high arousal threshold, airway dilator muscle responsiveness, and overnight rostral fluid shift [11].

This narrative review discussed the various treatment modalities for OSA and explained the multidisciplinary patient-centered approach that may improve patient participation and satisfaction.

## 2. Interventions for OSA

The initial treatment for OSA prior to the 1980s was a tracheostomy, which bypasses the impeded site of the upper airway, and unobstructed breathing occurs. However, this procedure has potential peri/postoperative complications, including mucous plugging, infection, tube dislodgement, disfigurement, communication challenges, and displacement [12].

In 1981, continuous positive airway pressure (CPAP) therapy was introduced by Collin Sullivan as an OSA treatment that provides a pneumatic splint for the nasopharyngeal airway to prevent airway collapse and improve oxygenation [13]. The 2015 American Academy of Sleep Medicine guidelines recommends using oral devices in cases where CPAP is contraindicated or alternative therapy is needed [2]. Despite the evolution of OSA treatment, PAP remains the preferred treatment with the most significant evidence of AHI and symptom reduction. Many other different PAP options are now available based on the mode of positive air pressure delivery and setting, such as auto-titrating positive airway pressure (APAP) and bi-level positive airway pressure (BIPAP) [14]. The individual must wear them regularly during sleep to avoid apneic events. However, many patients cannot tolerate PAP machines, resulting in poor adherence [15]. To improve adherence, advances have been made to monitor and track compliance and identify the challenge as leaks and residual apnea, with the data being downloaded via a smart card or cloud base [16]. Moreover, to maximize the benefits, the insurance providers are now mandating compliance of at least 4 h of usage for 70% of the night during the first month for continued device coverage [17].

Treatment for OSA should be tailored and offered to each patient so that each one will be able to receive the best treatment option. In general, CPAP is recommended as the first-line treatment for adult OSA. In patients with OSA who cannot tolerate the fixed pressure from the CPAP, APAP can be used. In patients who require a higher pressure than the maximal pressure of the CPAP machine to eliminate the apnea/hypopnea event, BIPAP is needed. For OSA with other comorbidities such as obesity hypoventilation syndrome, BIPAP may be considered for better hypercapnia improvement. Mandibular advancement devices can be an effective treatment, especially in those with small upper airway lumen and mild–moderate positional OSA [18]. Moreover, positional therapy may be used as an alternative treatment for mild positional OSA [19]. Inferior turbinate reduction and septoplasty have been shown to facilitate the OSA treatment in a patient with inferior turbinate hypertrophy and deviated nasal septum septoplasty, respectively. Maxillary/mandibular advancement can be considered in patients with maxillary/mandibular hypoplasia. For those with poor upper airway muscle function, nerve stimulation and myofunctional therapy can help increase muscle function. Hypoglossal nerve stimulation is found to be useful in patients with moderate to severe OSA who have no complete concentric collapse of the airway [20]. Recently, the electrical stimulation device to strengthen the tongue muscle tone has been shown to reduce AHI in mild OSA [21]. Pharmacotherapy may be added in those with high loop gain phenotypes to help decrease the plant and/or controller gain. Sedatives such as trazodone and eszopiclone have been found to be able to reduce AHI in patients with a low arousal threshold phenotype [22]. Unlike adults, adenotonsillectomy is considered the first-line therapy for pediatric OSA due to the risk of facial growth alteration from PAP mask usage [23,24]. In pediatric cases with high-arched palate or maxillary constriction, rapid maxillary expansion can be an alternative treatment [25].

Treatment for OSA may not be a one-size-fits-all problem due to the diversity of OSA. Therefore, many surgical interventions act as a salvage treatment after PAP treatment intolerance/failure, particularly in patients with surgically correctable abnormalities. This includes nasal polypectomy, adenoidectomy, tonsillectomy, uvulopalatopharyngoplasty, glossectomy, tongue base reduction, mandibular advancement, genioglossal advancement, hyoid myotomy suspension, maxillomandibular advancement, and bariatric surgery. Adenotonsillectomy in the pediatric population with OSA showed trivial results [26]. A recent meta-analysis by Bonetti et al. concluded that patients with pre-existing temporomandibular joints could be treated for OSA with mandibular advancement devices and will not experience a significant exacerbation of symptoms [27], disregarding TMJ disorders as a contraindication for MAD in OSA management. The most recent nonanatomic surgical option, hypoglossal nerve stimulator implantation, showed a surgical response in 75% of participants in a 5-year clinical trial. A surgical response in this trial was defined by the decrease in the apnea-hypopnea index (AHI) by more than 50% from the baseline and less than 20 events per hour [28]. Other therapeutic methods can also be considered an alternative or adjunctive treatment in mild–moderate OSA, such as oral appliances for patients with malocclusion, myofunction therapy [29], weight loss, and behavioral change, expiratory positive airway pressure (EPAP) therapy to help in preventing airway collapse during sleep. AHI increases with a supine position in more than half of the patients with OSA [30]. Therefore, positional therapy has been implemented as an additional treatment [19]. A recent meta-analysis concluded that treatment with trial and definitive MAD improved PSG parameters, with a statistically significant decrease in AHI (11.46 ± 9.65, *p* < 0.0001) and ODI (9.10 ± 8.47, *p* < 0.0016) with the definitive device [31]. Oral appliances can, therefore, be used as cheap and easy-to-use alternatives for OSA in the future [32]. Details are shown in Table 1.

## 3. Compliance with OSA Treatment

Despite the medical and surgical advancements, low compliance to the commonly used interventions, such as the PAP therapy, poses a significant challenge to patients. Adherence to CPAP machines varies between 30% and 80%, with 50% of the patients discontinuing therapy within the first year [33]. Social and economic factors decrease patient adherence to the intervention. On average, a CPAP machine costs over USD 250+, while the more complicated BiPAP machines cost USD 3000 [34]. A study by Park et al. concluded that 43% of the patients in the non-compliant group complained of physical discomfort with CPAP machines. Moreover, studies have explored an association between patient characteristics, such as age, sex, race, marital status, and obesity, with CPAP and OAT non-adherence. A recent meta-analysis [35] concluded a weak association; however, obesity remains the greatest risk factor for OSA [36]. Results from a longitudinal study [37] concluded that overweight and obese American adults had a severe apnea-hypopnea index (AHI). Underlying comorbidities such as diabetes mellitus type 2 (T2DM), hypercholesterolemia, and hypertension are strongly correlated with obesity and, therefore, OSA.

Lacking awareness of the OSA significance, enthusiasm for treatment availability, and equipment-related problem knowledge can be the potential barrier. Studies exploring patients’ readiness and cooperation suggest that patients who are confident and fully understand the OSA treatment plan are likely to adhere, making psycho-social well-being a key predictor of non-compliance [38]. Patients should be educated about the importance of OSA treatment that can help improve their symptoms, control their co-morbidities related to OSA, and avoid morbidity and mortality. Although healthcare professionals enhance adherence to OSA treatments, their lack of experience, knowledge, and communication skills contributes to mistrust in patients [39].

## 4. Individualized Care Approach to OSA Management

The Picker Foundation [40] was revolutionary in laying a framework for individualized care (personalized medicine and patient-centered approach) and increased patient–physician communication. Of these, a patient-centered approach, defined as promoting patient engagement and decision making, remains the epicenter of improving compliance with OSA interventions [2]. A recent study by Natsky et al. concluded that future care pathways for OSA must include a shorter waiting time for sleep study results and a sleep specialist to recommend a treatment [41], thus highlighting the need for patient inclusiveness. Patient engagement is vital for the treatment result. One method that improves patient satisfaction and clinical outcome is patient-centered care [42]. Patient-centered care covers compassion, respect, responsiveness to patients’ preferences, care coordination, information, physical/emotional support, and family and friends’ involvement. Patient-centeredness was included as one of the Six Aims, the performance for health care system improvement, proposed by the Institute of Medicine. These six main aims included were patient-centered, safe, timely, equitable, efficient, and effective. The patient-centered care approach should be prioritized to re-enforce the PAP acceptance, improving compliance.

A patient-centered care approach in OSA includes (1) identifying the problem and need of the patients, (2) patient readiness evaluation, (3) shared decision-making decisions, (4) promoting health-literacy education about the importance of OSA and its treatment, (4) patient engagement encouragement, (5) continuity of care improvement, and (6) PAP adherence and the outcome of the care assessment [15,16]. Barriers to the patient utilizing the therapy should be addressed. The open-ended question method needs to be utilized to identify underlying barriers and encourage the patient to come up with the expected outcome and thus have the buy-in from the patient [43].

In addition, personalized medicine utilizes the clinical information of the patient and the biological characteristics to design the therapies for the person, such as addressing and treating insomnia for the patient with an insomnia phenotype or addressing the cardiovascular issues in a patient with cardiovascular disease [2,44,45].

Another approach can be person-centered care. This model is emerging rapidly and has gone towards the team-based approach to addressing patient illness. This helps the interaction between the patient and the team and designing case-management based approach where a point person is assigned who works as a care coordinator between the patient and several other team members [46].

A recent randomized study by Naz et al. studied how to improve the CPAP program, which comprised interactive education, peer coaching, hands-on experience, and a motivational interview to the caregiver. Thus, a supportive environment along with shared decision making is an emerging approach to newly diagnosed OSA [47]. Despite these advancements, shared decision making by phenotyping is prone to disagreements between the patient and the doctor, which could be further limited by communication skills [48].

Since the PAP machine is used for a minimum of 4 h per night, the patient needs a behavioral commitment to follow a daily or a weekly routine. To ensure the patient’s readiness with the treatment choice, the patient must be provided with adequate experience to increase self-confidence [2]. Similarly, without an in-depth understanding of the PAP machine use, a poor understanding of the risk benefits can lead to misinformation and decreased confidence. A study by Parthasarathy et al. concluded that patient satisfaction (OR 4.6; 95% CI 2.3–9.3) significantly affected the primary care physician’s knowledge, while a certified physician was considered to deliver authentic information as opposed to a non-certified physician [49]. The sleep study report can also be patient-friendly by limiting technical information and using illustrations to convey key points such as oxygen saturation and the apnea-hypopnea index [50]. While supportive care from family and peers is essential to continue therapy, a review from 2020 concluded that behavioral therapy contributed to a 1.31 h/night (95% CI 0.95 to 1.66) increase in device usage [51].

Patients are often unable to recall or understand the treatment options for OSA and, therefore, do no follow-up with their physician, leading to an untreated OSA. Patients may also choose pharmacies and diagnostic challenges in selected cases and treatment recommendations [41,51], potentiating the use of patient decision aids, which familiarizes the patient with all available diagnostic and treatment options. Decide2Rest is one such web-based program for adults with newly diagnosed OSA, allowing them to make informed decisions and collaborate with the physician for future treatment changes [16]. While this has played a vital role in promoting nonadherence to OSA, patient engagement can be kept to a maximum with the use of smartwatches and wearable sleep-monitoring devices paired with applications on smartphones that monitor sleep schedules and oxygen saturation. Additionally, newer PAP devices record data on usage, efficacy, and leak, which is available to the patient, and keeps them engaged. This has resulted in increased PAP adherence in one week (6.3 ± 2.5 h) in individuals who had access to information compared with patients who did not (4.7 ± 3.3 h) [16]. In the era of telemedicine and online web portals that provide adequate information, patients can limit unnecessary visits to sleep centers. Table 2 summarizes the patient-centered approaches to OSA treatment.

In addition, technological advancements such as cloud bases and digital technologies can improve patient compliance and coaching. Moreover, attempts must be made to address the mask fit issue using 3-D facial scanning and 3-D printing to have a personalized mask rather than a one-shape fits all, predicting improved compliance [52].

## 5. Conclusions

Just five decades ago, sleep apnea was an unrecognized condition now being diagnosed and recognized as one of the most prevalent health care conditions with high morbidity and enormous health care burden with significant safety risk of MVA and work-related injuries among untreated patients. Since the inception of the CPAP mask four decades ago, advances have been made to improve the mask technology, which can help patients select from various masks to ensure fitness and compliance. However, CPAP therapy compliance remains very poor despite mask technology advancements and insurance providers mandating compliance for coverage. It is time to change our approach and move towards Individualized care, encompassing personalized medicine that is patient- and person-centered. This can help us save lives and costs. More research is needed in sleep medicine to understand the complex individual behavior and approach to combat them.

## Figures and Tables

**Table 1 medicina-58-01338-t001:** Surgical and Non-surgical Intervention of OSA.

Non-Surgical Intervention	Surgical Treatment
CPAP, BIPAP, APAP EPAP Lifestyle modifications (weight loss, exercise) Myofunctional therapy Oral appliances Pharmacological treatment	Hypoglossal nerve stimulation Genioglossal stimulation Surgical correction of anatomical abnormalities (tonsillectomy, nasal polypectomy, adenoidectomy, uvulopalatopharyngoplasty, glossectomy, tongue base reduction, mandibular advancement, genioglossal advancement, hyoid myotomy suspension, maxillomandibular advancement) Bariatric surgery Tracheostomy

**Table 2 medicina-58-01338-t002:** Patient-centric approaches to Obstructive Sleep Apnea treatment.

Patient’s readiness and beliefs	Socioeconomic Factors Patient’s expectations and compliance with the treatment
Health Literacy	Availability of specialists OSA treatment awareness programs Behavioral and Motivational Enhancement therapy Peer and Family Support
Shared Decision Making	Decision Aids Improved patient–physician interaction
Patient Engagement	Web-based portals Wearables and smartwatches technology Support groups

## Data Availability

Not applicable.

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
