# Peer review of "Patient-Centered Therapy for Obstructive Sleep Apnea: A Review"

_medicina, 2022, doi:10.3390/medicina58101338_

Round 1
Reviewer 1 Report
Thank you for submitting the manuscript titled “ Patient-Centered Therapy for Obstructive Sleep Apnea”. The authors have identified very important aspect of health issue. Overall manuscript flows well with the topic however there are areas which needs further improvements.
Major:
1. The manuscript requires more information about sleep apnea treatment vs low adherence or usage in the patient group. Are there any reason for this, for example accessibility of education, equipment support, cost?
2. The manuscript should include more co morbid condition that leads to certain condition. Please discuss genetic, gender, sex, lifestyle, age bias information.
3. Agree with the patient centered approach, however, discuss the current success model, limitation and future projection and its success likelihood.
4. Also, discuss any advancement in the technique itself, for example recently, CPAP machines industry are coming up with more user friendly to avoid the sleepless night.
Minor:
1. Many typo, spacing and font error identified through out the manuscript. Please proof read. Some examples.
Line 28, extra dot at the start.
Line 106-108- extra spacing
Line 139, 140 no comma in between the number points
Line 145, and 146 extra spacing
Table 1- Different font style
Table 2- Seems different font style and extra spacing
Author Response
Reviewer 1:
Thank you for submitting the manuscript titled “ Patient-Centered Therapy for Obstructive Sleep Apnea”. The authors have identified very important aspects of health issues. Overall manuscript flows well with the topic however there are areas which need further improvements.
Major:
Reviewer Comments:
- 1. The manuscript requires more information about sleep apnea treatment vs low adherence or usage in the patient group. Are there any reason for this, for example accessibility of education, equipment support, cost?
- The manuscript should include more co morbid condition that leads to certain condition. Please discuss genetic, gender, sex, lifestyle, age bias information.
Response: Thank you for your comments. The manuscript has been edited to contain the following information:
Compliance with OSA treatment
Despite the medical and surgical advancements, low compliance to the commonly used interventions, such as the PAP therapy poses a significant challenge to patients. Adherence to CPAP machines varies between 30% to 80%, with 50% of the patients discontinuing therapy within the first year31. Social and economic factors decrease patient adherence to the intervention. On average, a CPAP machine costs over USD 250+, while the more complicated BiPAP machines cost USD 300032. A study by Park et al. concluded that 43% of the patients in the non-compliant group complained of physical discomfort with CPAP machines. Moreover, studies have explored an association between patient characteristics, such as age, sex, race, marital status, and obesity with CPAP and OAT non-adherence. A recent meta-analysis33 concluded a weak association; however, obesity remains the greatest risk factor of OSA34. Results from a longitudinal study35 concluded that overweight and obese American adults had severe apnea-hypopnea index (AHI). Underlying comorbidities such as diabetes mellitus type 2 (T2DM), hypercholesterolemia, and hypertension are strongly correlated with obesity, and therefore, OSA.
Lacking awareness of the OSA significance, enthusiasm for treatment availability, and equipment-related problem knowledge can be the potential barrier. Studies exploring patients’ readiness and cooperation suggest that patients who are confident and fully understand the OSA treatment plan are likely to adhere, making psycho-social well-being a key predictor of non-compliance36. Patients should be educated about the importance of OSA treatment that can help improve their symptoms, control their co-morbidities related to OSA, and avoid morbidity and mortality. Although healthcare professionals enhance adherence to OSA treatments, their lack of experience, knowledge, and communication skills contributes to mistrust in patients37.
Reviewer Comments: Agree with the patient centered approach, however, discuss the current success model, limitation and future projection and its success likelihood. Also, discuss any advancement in the technique itself, for example recently, CPAP machines industry are coming up with more user friendly to avoid the sleepless night.
Response: Thank you for your comment. The manuscript now contains the following:
A recent randomized study by Naz et al. studied how to improve the CPAP program, which comprised of interactive education, peer coaching, hands-on experience, and a motivational interview with the caregiver. Thus, a supportive environment along with shared-decision making is an emerging approach to newly diagnosed OSA45. Despite these advancements, shared-decision making by phenotyping is prone to disagreements between the patient and the doctor, which could be further limited by communication skills46.
While this has played a vital role in promoting nonadherence to OSA, patient engagement can be kept to a maximum with the use of smartwatches and wearable sleep-monitoring devices paired with applications on smartphones that monitor sleep schedules and oxygen saturation. Additionally, newer PAP devices record data on usage, efficacy, and leak which is available to patients, and keeps them engaged. This has resulted in increased PAP adherence at one week (6.3 ± 2.5 h) in individuals who had access to information compared with patients who did not (4.7 ± 3.3 h)16. In the era of telemedicine and online web portals that provide adequate information, patients can limit unnecessary visits to sleep centers.
Reviewer Comments: Many typo, spacing and font error identified through out the manuscript. Please proof read. Some examples.
Line 28, extra dot at the start.
Line 106-108- extra spacing
Line 139, 140 no comma in between the number points
Line 145, and 146 extra spacing
Table 1- Different font style
Table 2- Seems different font style and extra spacing
Response: Thank you for pointing this out. We have corrected the above-mentioned errors in the manuscript.

Reviewer 2 Report
Dear Authors,
thank you for your submission.
Your paper concerns the interesting field of the patient-centered care in OSA patients.
Please correct the title to give the reader the information that in a narrative review.
Please do these corrections:
line 41 substitute the word abnormalities with morphology
line 50 substitute the word abnormalities with anomalies and please explain better which ones
line 51 OSA risks because of an increase in neck adipose tissue leading to a 51 high propensity of pharyngeal collapse[8]. This is not the unique mechanism connected with obesity
line 54 levator palatini and tensor palatini levator veli palatini, tensor veli palatini
at the end of the introduction specify the aim of the study
line 94 A recent meta-analysis by Bonetti et al the Authors declare the exact opposite The analysis of scientific literature evaluating the effects of MADs on TMD in patients with and without pre-existing signs and symp- toms showed that there is a moderate to low quality evidence that MAD therapy is not a risk factor for TMD signs and symptoms. Therefore, the presence of a TMD should not be considered a routine contraindication for the use of MADs in the management of OSA.
line 101 as oral appliances for patients with malocclusion Oral appliance is not only for patients with malocclusion. Cite
Segù M, Cosi A, Santagostini A, Scribante A. Efficacy of a Trial Oral Appliance in OSAS Management: A New Protocol to Recognize Responder/Nonresponder Patients. Int J Dent. 2021 Jun 17;2021:8811700. doi: 10.1155/2021/8811700. PMID: 34221017; PMCID: PMC8225417.
Levrini L, Sacchi F, Milano F, Polimeni A, Cozza P, Bernkopf E, Segù M; Italian dentist work group about OSAS Collaborators, Zucconi M, Vicini C, Brunello E. Italian recommendations on dental support in the treatment of adult obstructive sleep apnea syndrome (OSAS). Ann Stomatol (Roma). 2016 Feb 12;6(3-4):81-6. doi: 10.11138/ads/2015.6.3.081. PMID: 26941893; PMCID: PMC4755685.
Individualized Care Approach to OSA management
This approach could be very useful to improve the CPAP adherence but more to choose the best treatment for each patient.
In this chapter you completely forget all the other treatment modalities that you cited before that could be the best treatment for one patient.
Please address these indications.
Author Response
Reviewer 2:
Reviewer comment: Please correct the title to give the reader the information that in a narrative review.
Response: Thank you for your comment. The title has been edited to the following:
“PATIENT-CENTERED THERAPY FOR OBSTRUCTIVE SLEEP APNEA: A REVIEW.’
Reviewer comment: Please do these corrections: line 41 substitute the word abnormalities with morphology
Response: Thank you for pointing this out. The word has been replaced with morphology.
Reviewer comment: line 50 substitute the word abnormalities with anomalies and please explain better which ones.
Response: Thank you for pointing this out. It has now been edited to contain the following text:
Structural or physiological pathways approach can be taken in managing patients with OSA. Structural or anatomical impairment in OSA includes the main risk factor, obesity and craniofacial anomalies, such as bilateral mandibular hypoplasia and craniofacial microsomia7.
Reviewer comment: line 51 OSA risks because of an increase in neck adipose tissue leading to a 51 high propensity of pharyngeal collapse[8]. This is not the unique mechanism connected with obesity.
Response: Thank you so much for pointing this out. The manuscript has been edited to describe the well-known phenomenon of obesity-related OSA:
Obesity and large neck circumference are well-known OSA risks because of an increase in neck adipose tissue leading to a high propensity of pharyngeal collapse8
Reviewer comment: line 54 levator palatini and tensor palatini levator veli palatini, tensor veli palatini
Response: Thank you so much for pointing this out. The above-mentioned errors have been corrected.
Reviewer Comment: at the end of the introduction specify the aim of the study
Response: Thank you for your comment. The manuscript now contains the following text:
This narrative review discussed the various treatment modalities for OSA and explains the multidisciplinary patient-centered approach that may improve patient participation and satisfaction.
Reviewers Comment: line 94 A recent meta-analysis by Bonetti et al the Authors declare the exact opposite The analysis of scientific literature evaluating the effects of MADs on TMD in patients with and without pre-existing signs and symp- toms showed that there is a moderate to low quality evidence that MAD therapy is not a risk factor for TMD signs and symptoms. Therefore, the presence of a TMD should not be considered a routine contraindication for the use of MADs in the management of OSA.
Response: Thank you for pointing this out. We have corrected the manuscript, as follows:
A recent meta-analysis by Bonetti et al. concluded that patients with pre-existing temporomandibular joints could be treated for OSA with mandibular advancement devices and will not experience a significant exacerbation of symptoms27, disregarding TMJ disorders as a contraindication for MAD in OSA management.
Reviewer Comments: line 101 as oral appliances for patients with malocclusion Oral appliance is not only for patients with malocclusion. Cite
Segù M, Cosi A, Santagostini A, Scribante A. Efficacy of a Trial Oral Appliance in OSAS Management: A New Protocol to Recognize Responder/Nonresponder Patients. Int J Dent. 2021 Jun 17;2021:8811700. doi: 10.1155/2021/8811700. PMID: 34221017; PMCID: PMC8225417.
Levrini L, Sacchi F, Milano F, Polimeni A, Cozza P, Bernkopf E, Segù M; Italian dentist work group about OSAS Collaborators, Zucconi M, Vicini C, Brunello E. Italian recommendations on dental support in the treatment of adult obstructive sleep apnea syndrome (OSAS). Ann Stomatol (Roma). 2016 Feb 12;6(3-4):81-6. doi: 10.11138/ads/2015.6.3.081. PMID: 26941893; PMCID: PMC4755685.
Response: Thank you for your kind suggestions: The manuscript has been edited as follows:
A recent meta-analysis concluded that treatment with trial and definitive MAD improved PSG parameters, with a statistically significant decrease in AHI (11.46 ± 9.65, p < 0.0001) and ODI (9.10 ± 8.47, p < 0.0016) with the definitive device31. Oral appliances can therefore be used as cheap and easy-to-use alternatives for OSA in the future32.
Reviewer Comment: Individualized care approach to OSA Management.
Response: We appreciate the reviewer’s comment and the following para has been added in addition, this has been discussed interspersed throughout the manuscript
Treatment for OSA should be tailored and offered to each patient so that each one will be able to receive the best treatment option. In general, CPAP is recommended as the first-line treatment for adult OSA. In patients with OSA who cannot tolerate the fixed pressure from the CPAP, APAP can be used. In patients who require higher pressure than the maximal pressure of CPAP machine to eliminate the apnea/hypopnea event, BIPAP is needed. OSA with other comorbidities such as obesity hypoventilation syndrome, BIPAP may be considered for better hypercapnia improvement. Mandibular advancement devices can be an effective treatment, especially in those with small upper airway lumen and mild-moderate positional OSA18. Moreover, positional therapy may be used as an alternative treatment for mild positional OSA19. Inferior turbinate reduction and septoplasty have been shown to facilitate the OSA treatment in a patient with inferior turbinate hypertrophy and deviated nasal septum septoplasty, respectively. Maxillary/mandibular advancement can be considered in patients with maxillary/mandibular hypoplasia. For those with poor upper airway muscle function, nerve stimulation and myofunctional therapy can help increase muscle function. Hypoglossal nerve stimulation is found to be useful in patients with moderate to severe OSA who have no complete concentric collapse of the airway20. Recently, the electrical stimulation device to strengthen the tongue muscle tone has been shown to reduce AHI in mild OSA21. Pharmacotherapy may be added in those with high-loop gain phenotype to help decrease the plant and/or controller gain. Sedatives such as trazodone and eszopiclone have been found to be able to reduce AHI in patients with low arousal threshold phenotype22. Unlike adults, adenotonsillectomy is considered the first-line therapy for pediatric OSA due to the risk of facial growth alteration from PAP mask usage23,24. In pediatric cases with high-arched palate or maxillary constriction, rapid maxillary expansion can be an alternative treatment25.

Round 2
Reviewer 1 Report
Hello,
Thank you for addressing the comments. No further correction needed from my side.
Reviewer 2 Report
Dear Authors,
thank you very much in your big effort in addressing all my observations.
I accept the paper in the present form.